# SHARCS: SHARed Concept Space for Explainable Multimodal Learning

## Abstract

Multimodal learning is an essential paradigm for addressing complex real-world problems, where individual data modalities are typically insufficient for accurately solving a given modelling task. While various deep learning approaches have successfully addressed these challenges, their reasoning process is often opaque; limiting the capabilities for a principled explainable cross-modal analysis and any domain-expert intervention. In this paper, we introduce SHARCS (SHARed Concept Space) – a novel concept-based approach for explainable multimodal learning. SHARCS learns and maps interpretable concepts from different heterogeneous modalities into a single unified concept-manifold, which leads to an intuitive projection of semantically similar cross-modal concepts. We demonstrate that such an approach can lead to inherently explainable task predictions while also improving downstream predictive performance. Moreover, we show that SHARCS can operate and significantly outperform other approaches in practically significant scenarios, such as retrieval of missing modalities and cross-modal explanations. Our approach is model agnostic and easily applicable to different types (and number) of modalities, thus advancing the development of effective, interpretable, and trustworthy multimodal approaches.

## 1 Introduction

Multimodal learning has emerged as a critical research area due to the need for AI systems that can effectively handle complex real-world problems where individual modalities are insufficient to solve a given task. Moreover, in safety-critical domains such as biology and transportation, it is vital to develop interpretable and interactive multimodal agents that can explain their actions and interact with human experts effectively (Rudin, 2019; Shen, 2022). This presents a unique challenge for researchers and a crucial step for deploying effective and trustworthy multimodal agents.

Existing deep learning (DL) systems for multimodal learning attain high performance by blending information from different data sources (Radford et al., 2021; Lei et al., 2021). However, the opaque reasoning of DL models (Rudin, 2019) hinders the human ability to draw meaningful connections between the modalities, which could potentially lead to novel insights and discoveries. To address this issue, many self-explainable methods were released (Koh et al., 2020; Zarlenga et al., 2021; Alvarez-Melis & Jaakkola, 2018; Chen et al., 2019), offering an effective solution to bridge this knowledge gap. These methods can extract intuitive and human-readable explanations, and some even facilitate interaction with human experts, enabling a deeper understanding of the problem. However, they are often limited to single data modalities. A recent line of research focuses explicitly on developing or adapting existing methods for multimodal settings (Rodis et al., 2023). While relevant, they are typically tailored for specific multimodal scenarios (Wang et al., 2015), provide only local explanations (Park et al., 2018; Li et al., 2018) or generate explanations for just one of the modalities (Kim et al., 2018b) using an extra modality, thus failing to provide a general solution to multimodal problems.

In this paper, we introduce SHARCS (SHARed Concept Space), a novel interpretable concept-based approach (described in Section 2) designed to address general multimodal tasks. Our experiments (Section 3) demonstrate on four common data modalities (tabular, text, image, and graph data) that SHARCS (i) outperforms unimodal models and matches the task performance of existing baselines on challenging multimodal settings, (ii) attains high task accuracy even when a modality is missing,

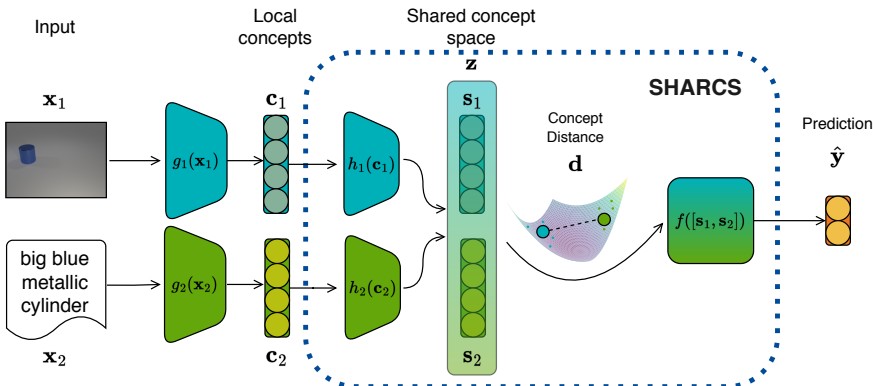

Figure 1: **SHARCS (SHARed Concept Space)**: for each modality $i$, the concept encoder module $g_i$ produces a local concept embedding $\mathbf{c}_i$. SHARCS then maps local concept embeddings into a shared concept representation $\mathbf{s}_i$. To generate a semantically meaningful shared space, SHARCS minimises the distance between shared concepts of similar objects from different modalities. Finally, the label predictor $f$ takes as input the concatenation of all shared concepts $\mathbf{s_i}$ to solve the task at hand.

(iii) generates intuitive concept-based explanations for task predictions, and (iv) generates simple concept-based explanations for a data modality using the concepts emerging from other modalities, allowing human experts to uncover hidden cross-modal connections.

Our contributions can be summarised as follows:

- We introduce SHARCS – a novel concept-based approach for explainable multimodal learning. SHARCS is a model-agnostic approach that learns and maps interpretable concepts from different heterogeneous modalities into a unified shared concept space, which leads to an intuitive projection of semantically similar cross-modal concepts.
- We show that SHARCS is able to outperform a variety of unimodal and multimodal baselines on four different tasks, especially in scenarios with missing modalities;
- We demonstrate the interpretable capabilities of SHARCS, providing valuable insights of the cross-modal relationship between the different modalities at hand.

## 2 SHARCS: SHARED CONCEPTS SPACE

### 2.1 PRELIMINARIES

Multimodal learning systems process data set of $i = 1, \ldots, n \in \mathbb{N}$ modalitis, each describing an input sample $m \in \mathbb{N}$ in potentially heterogeneous feature spaces $\mathbf{x}_{im} \in X_i \subseteq R^d$ with dimensionality $d \in \mathbb{N}$. In supervised settings, data sets may provide global (or modality-specific) sets of $l \in \mathbb{N}$ task labels $\mathbf{y}_m \in Y \subseteq R^l$ ($\mathbf{y}_{im} \in Y_i \subseteq R^l$). In these scenarios, multimodal learning systems are trained to map inputs from feature spaces $X_1, \ldots, X_n$ to the output task spaces $Y_1, \ldots, Y_n$.

### 2.2 ARCHITECTURE

SHARCS combines the information extracted from different data modalities during training. In particular, it aims at joining high-level human-interpretable concept representations (as defined by Ghorbani et al. (2019)) in a shared concept space rather than joining standard embeddings, which are typically uninterpretable (Rudin, 2019). SHARCS first extracts a set of $k \in \mathbb{N}$ concepts from individual modalities and then combines them as multimodal concept representations in a shared concept space $S \subseteq [0, 1]^t$ of size $t \in \mathbb{N}$. This renders intuitive concept-based explanations and allows human experts to interact with the learnt concepts to gain insights into the mutual relationships between data modalities. For instance, in SHARCS, a red ball is represented by a multimodal concept whose representation in the shared space is invariant w.r.t. the input modality (e.g., image, text, etc.).

**Local concepts** Figure 1 describes an example of SHARCS applied on two data modalities. The first part of the model is composed of a set of distinct concept encoder functions $g_1, \ldots g_n$, one for each

modality $i = 1, \ldots, n \in \mathbb{N}$. The concept encoder function $g_i : X_i \to \mathbb{R}^k$ maps inputs from the $i$-th modality to the set of (local) concepts available for that modality. In practice, we instantiate a concept encoder using a modality-specific architecture e.g., a set of feed-forward layers for tabular data, convolutional layers for images, or message passing layers for graphs. Modality-specific architectures $\phi_1, \ldots, \phi_n$ map inputs to latent concept representations i.e., $\phi_i : X_i \to \mathbb{R}^k$. A concept encoder then maps latent concept representations into a local concept space using a batch scaling $\circledast : \mathbb{R}^{b \times k} \to \mathbb{R}^k$ (where $b \in \mathbb{N}$ is the batch size) and a sigmoid activation function $\sigma : \mathbb{R} \to [0, 1]$ i.e., $g_i = \sigma \circ \circledast \circ \phi_i$. To make the model focus on relevant concepts, we perform batch rescaling before applying the sigmoid activation. This ensures that an input sample $m$ activates a concept only if the concept representation, prior to rescaling, significantly differs from the representations of other samples in the same batch:

$$\mathbf{c}_{im} = \left( 1 + \exp \left( - \left( \phi_i(\mathbf{x}_{im}) \underset{j \in B_{im}}{\circledast} \phi_i(\mathbf{x}_{ij}) \right) \right) \right)^{-1} \tag{1}$$

where $B_{im} \subseteq \mathbb{N}$ represents the indexes of the batch of samples including the $m$-th sample, $\circledast$ is a permutation-invariant batch rescaling function (such as batch normalisation), and $\mathbf{c}_i$ are the local concepts of the $i$-th modality.

**Shared concepts** SHARCS then maps the local concepts $\mathbf{c}_i$ into a shared concept space. To this end, SHARCS applies a modality-specific set of concept encoders $h_1, \ldots, h_n$ mapping local concepts $\mathbf{c}_i \in C \subseteq [0, 1]^k$ into a set of shared concept embeddings $\mathbf{s}_i \in S \subseteq [0, 1]^t$ of size $t \in \mathbb{N}$ i.e., $h_i : C_i \to S$. Shared concept encoders resemble the structure of local encoders applying batch rescaling and a sigmoid activation on top of learnable parametric functions $\psi_1, \ldots, \psi_n$:

$$\mathbf{s}_{im} = \left( 1 + \exp \left( - \left( \psi_i(\mathbf{c}_{im}) \underset{i=1,\ldots,n \,\wedge\, j \in B_{im}}{\circledast} \psi_i(\mathbf{c}_{ij}) \right) \right) \right)^{-1} \tag{2}$$

Thanks to this operation, our model blends information from different data modalities into the same space, enabling the generation of unified concept manifolds.

**Task prediction.** Finally, the model concatenates the shared concepts $\mathbf{s}_i$ from each modality and uses them to solve the task at hand. To solve the task, a label predictor function $f : S^n \to Y$ maps the shared concepts to a downstream task space $Y \subseteq \mathbb{R}^l$: $\hat{\mathbf{y}}_m = f(\mathbf{s}_{1m} | \ldots | \mathbf{s}_{nm})$ where the symbol $|$ represents the concatenation operation. We provide more details about SHARCS in Appendix A.

## 2.3 LEARNING PROCESS

The SHARCS architecture enables the integration of information from potentially diverse data modalities into a unified vector space. However, concept encoders may learn distinct concepts for different tasks, resulting in different concepts being mapped to the same region in the shared vector space. To avoid this, we generate a semantically homogeneous shared space by introducing an additional term in our model's loss function $\mathcal{L}$. By doing so, the model is encouraged to establish connections between concepts learned from different modalities, promoting a semantically coherent shared space:

$$\mathcal{L}(\mathbf{y}, \hat{\mathbf{y}}, \mathbf{s}) = \mathcal{T}(\mathbf{y}, \hat{\mathbf{y}}) + \frac{\lambda}{|M|} \sum_{(i,q) \in M \subseteq \binom{\{1,\ldots,n\}}{2}} ||\mathbf{s}_i - \mathbf{s}_q||_2 \tag{3}$$

where $\lambda \in \mathbb{R}$ is a hyperparameter that controls the strength of our semantic regularization, $\mathcal{T}$ is a task-specific loss function (such as cross-entropy), $M$ is a subset of all possible pairs of modalities $\binom{\{1,\ldots,n\}}{2}$, and $||\mathbf{s}_i - \mathbf{s}_q||_2$ represents the Euclidean norm between the shared representation of the same sample in the two different modalities $i$ and $q$. In our solution, we randomly draw samples to compute the semantic regularisation loss at every iteration. It is worth noting that a model based on SHARCS can also accommodate local tasks allowing the definition of modality-specific loss functions, which can be included in the global optimisation process (cf. Appendix A).

A model based on SHARCS offers three learning mechanisms enabling different forms of training according to concrete use cases:

- **End-to-end**: This method trains all model components together, allowing a joint optimisation of global (and local) tasks, local and shared concepts.

- **Sequential**: This method first trains the full model without SHARCS-specific components (i.e., $h_i$, $f$, and the SHARCS loss). After this pre-training phase, local models freeze, and SHARCS components begin training. This approach enables the independent development of local concepts preventing dominant modalities from overshadowing concepts of weaker modalities.
- **Local pre-training**: This method first trains modality-specific models only. After this local pre-training, local models freeze, and all the other components start their training.

In contrast to Concept Bottleneck Models (CBMs) (Koh et al., 2020), we do not use any supervision on local concepts, allowing the model to learn them directly from data.

## 2.4 MULTIMODAL CONCEPT-BASED EXPLANATIONS

**Unimodal and multimodal explanations.** The key advantage of SHARCS with respect to existing multimodal models is that it provides intuitive concept-based explanations. Similarly to unimodal unsupervised concept-based models (Ghorbani et al., 2019), we can use SHARCS to assign semantic meaning to concept labels by visualizing the "prototypes" of a concept, represented by the examples with the highest values (concept active) or with the lowest values (concept inactive) for that concept in the shared space. Thanks to the interpretable architecture, SHARCS does not require an external algorithm to find these samples as opposed to post-hoc methods such as Ghorbani et al. (2019). More formally, we can retrieve a prototype $\gamma_v \in \{0,1\}^t$ of a concept $v$ in a modality $i$ by taking the sample with the SHARCS shared concept representation with the highest (lowest) value across all the $s_i$ seen during training.

Moreover, SHARCS can also provide a semantic contextualisation for an input sample $m$ by visualising the input samples whose embeddings are closer in the shared space embeddings to the given input. More formally, given a reference modality $i$ we can identify the set of closest samples to the input $m$ in a radius $\rho \in \mathbb{R}$ as follows:

$$E = \{j \in B_{train} \mid ||\mathbf{s}_{im} - \mathbf{s}_{ij}||_2 < \rho\} \tag{4}$$

where $B_{train} \subseteq \mathbb{N}$ is the set of indexes of training samples, $E \subseteq B_{train}$ is the set of the closest samples in the shared space. This form of visualisation is often used to find relevant clusters of samples sharing some key characteristics. By showcasing examples from the input concept's family, the SHARCS allows users to comprehend why these examples are classified similarly.

Both can be applied to global concepts, as they are the concatenation of the shared concepts. The only difference it to consider $\mathbf{z}_j = (\mathbf{s}_{1j}|\dots|\mathbf{s}_{nj}) \in [0,1]^{n \times t}$, instead of $s_i$.

**Cross-modal explanations.** SHARCS offers unique forms of explanations which go significantly beyond simple unimodal interpretability. Indeed, SHARCS enables cross-modal explanations, allowing one modality to be explained using another. Specifically, we can use an input sample in a specific modality to retrieve the most similar examples from other modalities. To this end, we can select training samples from the other modalities, which are closer in the shared concept space to the sample $m$ being explained:

$$E = \{j \in B_{train}, \, q \in \{1, \dots, n\} \mid ||\mathbf{s}_{im} - \mathbf{s}_{qj}||_2 < \rho\} \tag{5}$$

As before, it is also possible to visualise how a concept $v$ of a modality $i$ is interpreted in the other modality $q$, visualising the samples with the highest (or lowest) value of the concept $v$ in modality $q$ across all $s_q v$. In such a manner, it is possible to translate a key feature from one modality to another.

These functionalities are particularly valuable when a modality's features are less human-interpretable than others. Visualizing the relationships between modalities enables cross-modal interpretability by emphasizing the semantic interconnections between concepts of different modalities.

**Inference with missing modalities.** Another unique feature of SHARCS is that the shared concept space enables it to process inputs with missing modalities effectively. Indeed, the original representation of an input $m$ of a missing modality $i$ can be effectively approximated using the shared concepts of another reference modality $q$. To this end, we just need to find the shared concept $\mathbf{s}_{ij}$ observed during training from the missing modality, which is closest to a shared concept of the reference modality $m' = \arg\min_{j \in B_{train}} ||\mathbf{s}_{qm} - \mathbf{s}_{ij}||_2$

This way we can approximate the missing shared concept representation $\mathbf{s}_{im}$ from the missing modality as follows:

$$\mathbf{s}_{im'} = \Big(1 + \exp\Big(-\Big(\psi_i(\mathbf{c}_{im'}) \underset{i=1,\ldots,n \ \wedge \ j \in B_{im'}}{\circledast} \psi_i(\mathbf{c}_{ij})\Big)\Big)\Big)^{-1} \approx \mathbf{s}_{im} \qquad (6)$$

## 3 Experiments

Our central hypothesis is that SHARCS allows for an efficient, accurate and interpretable multimodal learning. To address these aspects, we design our experiments along two main points: (**Multimodal generalisation performance**) Through a series of experiments, we first evaluate SHARCS' capabilities for multimodal learning in different practically-relevant scenarios. Then, we compare SHARCS performance to unimodal and multimodal baselines, some of which are not interpretable; (**Interpretability**) We qualitatively showcase SHARCS capabilities for learning semantically plausible, explainable and consistent (multimodal) concepts.

**Tasks and datasets.** We evaluate our hypotheses on four multimodal tasks, each leveraging a pair of multimodal datasets such as tabular, image, graph, and text data. The four multimodal, or global, tasks are designed such that the models need to leverage both modalities in order to provide correct predictions. Models that will learn only from one of the modalities will be able to solve a partial (local) single-modality task but will typically exhibit random performance on the global multimodal task.

(i) **"XOR-AND-XOR"** task: Combines tabular and graph data, each modeling a local XOR task. The dataset has 1000 samples for each modality. The tabular modality uses bit-strings (2 used for solving the 'xor' and 4 random), and the graph modality contains four types of graphs (each graph can be transformed to a binary 2-bit binary encoding). The global task is binary 'and' problem, combining the local 'xor' results. (ii) **"MNIST+Superpixels"** task: Consists of 60,000 pairs of MNIST images and superpixel-graphs. Local tasks involve image and graph classification, with the global task being the sum of the two digits. (iii) **"HalfMNIST"** task: Combines 60,000 image and graph samples, where each modality represents one half of the sample. The goal is MNIST classification. (iv) **"CLEVR"** task: Adapts the CLEVR benchmark for visual question answering by generating text captions for 8000 images. The multimodal task is binary, predicting whether the caption matches the image. Additional dataset details can be found in Appendix B.

**Modeling details.** As discussed earlier, SHARCS learns modality-specific concepts before combining them in a shared space. Therefore, since we consider tasks that combine different modalities, we use different models. Specifically: (i) for tabular data, we use a 2-layer Feed Forward Network; (ii) for images, a 2 layers CNN (MNIST+Superpixels, HalfMNIST) or a pre-trained ResNet18 (He et al., 2015) (CLEVR); (iii) for text, a 2-layer Feed Forward Network after computing the text representation with TF-IDF; and (iv) for graphs, 4 layers of GCN (Kipf & Welling, 2016) (XOR-AND-XOR) or 2 layers of Spline CNN (Fey et al., 2018) (MNIST+Superpixels, HalfMNIST). Appendix C provides further details about model compositions and used hyperparameters. Note that, since in this paper, we are focusing on evaluating the efficacy of SHARCS in a multimodal setting rather than pursuing state-of-the-art performance; all approaches use the same (local) backbone architectures. Nevertheless, as SHARCS is model agnostic, these can be easily extended to more sophisticated (but likely less efficient) architectures.

**Baselines and experiments.** We begin by examining SHARCS' multimodal capabilities. In our initial experiments, we compare SHARCS with models trained solely on single modalities. These unimodal models include both basic concept-less models and concept-based variations. In the subsequent experiments, we assess SHARCS' performance against several multimodal baseline models. These baselines consist of: (i) A standard multimodal approach called 'Simple Multimodal,' which combines uninterpretable embedded representations from individual local models (ii) A concept-based variant known as 'Concept Multimodal,' similar to the previous approach but additionally computes and uses local concepts without sharing them (iii) A 'Relative Representations' multimodal approach (Moschella et al., 2023), which constructs relative mapped representations of each sample in relation to a given anchor within a shared space. This approach requires a two-stage training process: first for building representations for each modality and then for mapping them in the shared relative space. Furthermore, we consider a practical multimodal scenario involving missing modalities. In this setup, we train multimodal models using both modalities, but during inference, one of the modalities is

Table 1: Accuracy (%) of SHARCS compared to non-interpretable (Simple Multimodal and Relative representation) and interpretable (Concept Multimodal) multimodal baselines. Generally, SHARCS achieves comparable performance than the other baselines.

| Model | XOR-AND-XOR Acc. | MNIST+SuperP. Acc. | HalfMNIST Acc. | CLEVR Acc. |
|---|---|---|---|---|
| Simple | $99.3 \pm 0.5$ | $86.6 \pm 3.0$ | $94.2 \pm 0.2$ | $59.5 \pm 9.5$ |
| Concept | $99.0 \pm 0.8$ | $88.2 \pm 0.1$ | $93.9 \pm 0.0$ | $90.1 \pm 1.0$ |
| Relative | $\mathbf{99.5} \pm 0.3$ | $80.4 \pm 0.2$ | $\mathbf{95.6} \pm 0.1$ | $48.7 \pm 0.5$ |
| SHARCS | $98.7 \pm 0.5$ | $\mathbf{89.6} \pm 0.1$ | $94.0 \pm 0.1$ | $\mathbf{90.2} \pm 0.2$ |

replaced with an auxiliary one. For example, instead of representing a six as an image and a four as a graph, we represent both a six and a four as images.

**Evaluation metrics.** We repeat each experiment several times (three times in the case of CLEVR and five times for the other three) and report a mean and standard error for each metric we use. Each model has been evaluated using test classification accuracy to evaluate multimodal generalisation performance. Furthermore, we also report the completeness score to quantitatively assess the concept quality (for SHARCS and Concept Multimodal). The completeness score assesses how the learnt concepts are suitable to solve the downstream task. To compute it, we train a decision tree, which takes the binarised global concepts at the input. To evaluate the performance of SHARCS to the ones of the 'Relative representation' and 'Concept Multimodal' variants in the missing modality settings, we compute their accuracy in this scenario.

## 4 RESULTS AND DISCUSSION

### 4.1 MULTIMODAL GENERALISATION PERFORMANCE

**SHARCS outperforms unimodal models.** As a proof-of-principle of our method, we benchmarked it against unimodal approaches (see Figure 8 in the Appendix D). SHARCS achieves good performance across all four multimodal tasks, consistently outperforming (up to 81%) the unimodal baselines. On tasks that are practically solvable using only one of the modalities, such as the case of HalfMNIST, SHARCS can outperform the other baselines by up to 18%. While this behaviour is expected, these experiments further validate our design decisions – that while for some problems, one modality may be sufficient, employing all available modalities can provide great benefits.

**SHARCS' generalisation is on par with non-interpretable multimodal models.** The results presented in Table 1 show that SHARCS achieves slightly better or comparable performance than the other multimodal baselines. In particular, our approach can maintain good performance, despite the bottleneck introduced for computing concepts and the constraint of the shared space. More importantly, both concept-based approaches are the only two that can accurately model the CLEVR task, which further justifies the utility of the concept embeddings.

**SHARCS can successfully handle missing modality situation.** Table 2 shows that SHARCS generalisation performance is consistently accurate in missing modality scenarios, significantly outperforming (in all but one case) the

Table 2: The performance of SHARCS (Accuracy (%)) in scenarios with missing modalities, compared to Relative representation and Concept Multimodal variants. The global task accuracy is presented as a reference. SHARCS performs better than the baselines, particularly on harder tasks requiring both modalities. In some scenarios, SHARCS is able to retrieve modalities, leading to better downstream performance than the original data.

| Dataset | Model | Missing Modality 1st Modality | 2nd Modality | Global Task Accuracy |
|---|---|---|---|---|
| XOR-AND-XOR | Relative | $80.1 \pm 6.4$ | $82.8 \pm 2.2$ | $\mathbf{99.5} \pm 0.3$ |
| | Concept | $68.0 \pm 2.0$ | $57.0 \pm 6.1$ | $99.0 \pm 0.8$ |
| | SHARCS | $\mathbf{98.6} \pm 0.9$ | $\mathbf{91.9} \pm 1.2$ | $98.7 \pm 0.5$ |
| MNIST+SuperP. | Relative | $52.6 \pm 4.9$ | $30.1 \pm 2.4$ | $80.4 \pm 0.2$ |
| | Concept | $13.7 \pm 3.9$ | $10.8 \pm 2.6$ | $88.2 \pm 0.1$ |
| | SHARCS | $\mathbf{98.0} \pm 0.0$ | $\mathbf{82.5} \pm 0.4$ | $\mathbf{89.6} \pm 0.1$ |
| HalfMNIST | Relative | $92.9 \pm 1.4$ | $\mathbf{60.1} \pm 3.4$ | $\mathbf{95.6} \pm 0.1$ |
| | Concept | $89.4 \pm 1.3$ | $13.4 \pm 2.1$ | $93.9 \pm 0.0$ |
| | SHARCS | $\mathbf{96.5} \pm 0.0$ | $55.1 \pm 3.0$ | $94.0 \pm 0.1$ |
| CLEVR | Relative | $49.9 \pm 0.0$ | $49.0 \pm 0.1$ | $48.7 \pm 0.5$ |
| | Concept | $51.4 \pm 2.8$ | $48.6 \pm 2.7$ | $90.1 \pm 1.0$ |
| | SHARCS | $\mathbf{93.1} \pm 0.6$ | $\mathbf{93.4} \pm 0.4$ | $90.2 \pm 0.2$ |

other baselines. We attribute this to SHARCS' ability to construct a better and less noisy concept space, where concepts from the different modalities provide a better representation of the samples, thus leading to more precise retrieval of the missing counterpart.

**SHARCS with a missing modality outperforms all the baselines and itself with both modalities.** Table 2 shows that, in situations where a modality is more expressive than the other (such as MNIST+Superpixels and CLEVR), SHARCS with a missing modality achieves higher performance than all the other baselines and itself using both modalities. We further analyse these capabilities of SHARCS by benchmarking its retrieval performance of samples with specific characterises. Furthermore, we performed an analysis on CLEVR, where we checked for each model which characteristics of the retrieved sample matched with the ones of the object used as the source (See Table 5 in the Appendix D). SHARCS can better match such (semantically meaningful) concepts than the other baselines. This analysis also shows that some characteristics are more challenging to learn than others (such as colour), which can have a negative effect on downstream performance. Nevertheless, SHARCS, being interpretable, allows for diagnosing and mitigating such issues, which can be highly beneficial in practical scenarios.

## 4.2    INTERPRETABILITY

**SHARCS discovers meaningful concepts.** SHARCS, as the Concept Multimodal baselines created by us, is capable of learning useful concepts from a task viewpoint, as evidenced by a completeness score close to the Accuracy in Table 3. In particular, SHARCS achieve a higher completeness score on three of the four datasets than the solution without shared space, with an improvement of up to 10% completeness score in MNIST+Superpixels. By employing a shared space, SHARCS is able to denoise some concepts using the other paired modality. By doing so, it is able to collapse some (likely unimportant) concepts into one. For instance, in the case of CLEVR, samples from the text modality labelled with "metal ball" can have a different concept representation from ones labelled with "shiny ball". Since SHARCS can efficiently leverage the other (image) modality, such concepts collapse into one - more semantically meaningful concept. This is supported by the number of shared concept representation clusters (a cluster of shared concept representation is composed by samples with the same binarised shared concept representation) found by each model. SHARCS reduce the number of clusters by up to 3.7 times. This provides more valuable and clearer global explanations, leading to fully interpretable results.

**SHARCS' concepts shed light on how the task can be solved.** As we showed in Section 4, SHARCS discovers concepts that are specifically important in solving the tasks. We can further qualitatively assess and visualise this property by employing a decision tree (used for computing the completeness score) on the prediction step, trained with the learnt concepts (Figure 2). In this manner, the user can understand how different concepts are employed in the decision process. Specifically, it allows insights into which are the most important concepts, understand better the task at hand, how different concepts combine and why specific samples are classified in a certain way.

**SHARCS explains one modality using the other.** Finally, we focus on the ability of SHARCS to retrieve and explain one modality by using the other. Specifically, we visualise examples from a modality by sampling the shared concept space using its pair. We retrieve such samples and visually compare them to samples retrieved by the 'Relative representation' and 'Concept Multimodal' variants. Here we present only results from the MNIST+Superpixels dataset and provide additional results and figures for the other datasets in Appendix D. The samples presented in Figure 3 show that SHARCS, in general, can accurately retrieve such samples, whilst the other two counterparts struggle and produce "random" retrievals. The same can be done on single concept values, illustrating how

Table 3: Accuracy (%) and Completeness Score (%) of SHARCS and Concept Multimodal. Generally, they achieves high completeness score compared to the accuracy.

| Model | XOR-AND-XOR | | MNIST+SuperP. | | HalfMNIST | | CLEVR | |
| | Acc. | Compl. | Acc. | Compl. | Acc. | Compl. | Acc. | Compl. |
| --- | --- | --- | --- | --- | --- | --- | --- | --- |
| Concept | $99.0 \pm 0.8$ | $96.2 \pm 1.2$ | $88.2 \pm 0.1$ | $78.9 \pm 1.4$ | $93.9 \pm 0.0$ | $91.3 \pm 0.1$ | $90.1 \pm 1.0$ | $82.3 \pm 1.2$ |
| SHARCS | $98.7 \pm 0.5$ | $\mathbf{98.0} \pm 1.2$ | $89.6 \pm 0.1$ | $\mathbf{88.7} \pm 0.2$ | $94.0 \pm 0.1$ | $\mathbf{92.6} \pm 0.3$ | $90.2 \pm 0.2$ | $81.5 \pm 1.1$ |

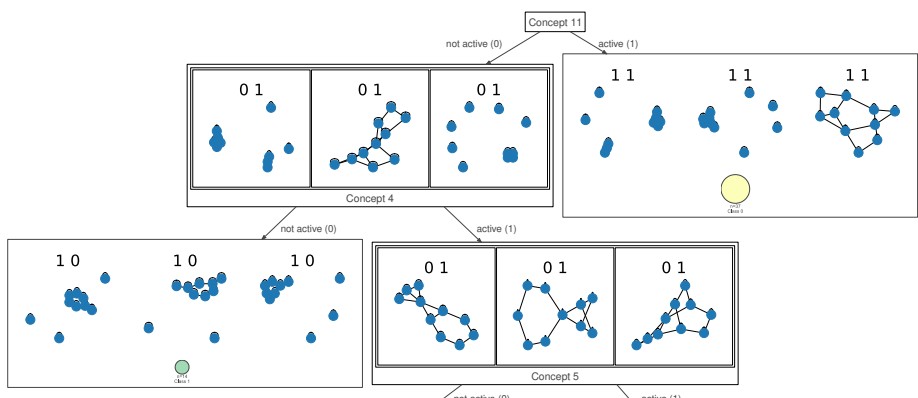

Figure 2: First layers of the decision tree visualisation of SHARCS concepts on the XOR-AND-XOR dataset. At every split, it shows the concept that is considered to make the decision, and it can be active (right branch) or non-active (left branch). If the node is not the roof, it also shows three samples with the highest (concept active) or the lowest (concept non-active) value for the concepts of the previous split, among the ones that respect all the previous split conditions. Each leaf shows the class distribution of the samples that it represents, in addition to the most characteristic samples.

one specific concept can be represented in the other modality. This can be done by retrieving the examples with the closest values in that specific dimension/concept.

**Different modalities are overlapped in the SHARCS' shared space.** Furthermore, Figure 3d depicts the shared space of the MNIST+Superpixels dataset created by SHARCS. Here, it is evident that similar examples from different modalities are mapped closer together. This, however, is expected, as in the objective function, we are minimising the distance between similar concepts. Nevertheless, we believe this to be an extremely useful property, especially in scenarios when modalities lack expressiveness (from a human perspective) or the commonalities between the modalities are much more nuanced. In such cases, our approach can elucidate important inner relationships between modalities (and samples), which is beneficial for many downstream applications in domains such as medicine, biology and healthcare.

## 5 RELATED WORK

**Multimodal Learning** SHARCS addresses multimodal learning by constructing a shared representation space from both modalities. As such, it is closely related to the method of (Moschella et al., 2023), which builds on relative representations. (Moschella et al., 2023) constructs $n$ models that are trained on a particular (unimodal) local task and a set of randomly selected samples from each modality, called anchors. The relative representation for a sample in a dataset $i$ is calculated by computing the relative distance between the representation given by the model $m_i$ for that sample

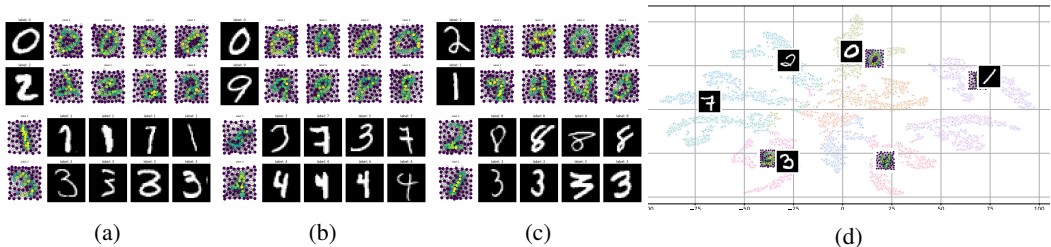

|        |        |        |        |
| :----: | :----: | :----: | :----: |
| (a)    | (b)    | (c)    | (d)    |

Figure 3: (a-c) Retrieval examples obtained by (a) SHARCS, (b) Relative representation, and (c) Concept Multimodal; on the MNIST+Superpixels dataset. The top two rows are samples of retrieved graphs using images, while the bottom two are retrieved images using graph samples. (d) tSNE plot of the SHARCS concept space

and that of each anchor of that specific modality. However, such an approach can quickly lose performance in scenarios where the two modalities differ significantly, which was also evident from our results. In contrast, SHARCS can alleviate these shortcomings by constraining the concepts in a shared space. In this context, our method is also related to contrastive multimodal approaches (Chen et al., 2020a), such as the one used in CLIP (Radford et al., 2021). Such approaches attempt to map cross-modal samples by minimising/maximising cosine similarity to similar/dissimilar samples from the two modalities. Although such methods are in principle applicable to any type and number of modalities, they have only been only tested in scenarios with fully available pairs, apart from having exponential complexity with respect to the number of modalities. Moreover, contrastive learning models suffer from the modality gap problem (Liang et al., 2022), whereas SHARCS is designed to reduce this gap.Our approach also relates to the methods used in visual question answering (Saqur & Narasimhan, 2020), where GNNs are used to perform multimodal fusion. While related these approaches are model/task-specific, whereas SHARCS is model agnostic and can be applied to any type and number of models and modalities. More broadly, SHARCS relates to several methods that address explainable multimodal learning (Rodis et al., 2023). However, in contrast to many methods in this area (Kim et al., 2018b), that use one modality to solve the task and another to produce explanations - SHARCS is more general and applicable to a wide range tasks, allowing explanations that can relate either specifically to a modality or explaining the cross-modal relationships.

**Concept-based Explanations ((Ghorbani et al., 2019))** Since we focus on interpretable representations in addition to multimodal learning, our work also relates to the class of concept-based models. Concept-based models are interpretable architectures that allow predictions to be mapped directly to human-understandable concepts, making the model's decision-making process transparent (Kim et al., 2018a; Chen et al., 2020b; Koh et al., 2020; Rudin, 2019; Shen, 2022). Our solution builds on these capabilities and follows CBM (Koh et al., 2020), which uses concept supervision to extract understandable explanations. To this end, however, these approaches have been designed for unimodal settings, underperforming when applied and evaluated on multimodal tasks (as discussed in Section 4). SHARCS attempts to overcome this challenge by extending these capabilities to local (unimodal) and global (multimodal) tasks. SHARCS can learn concepts that share common characheristics. This enables comprehensible explanations at different levels (modality-specific or global) and between different modalities - a unique and novel property of our method.

## 6  CONCLUSION

In this paper, we emphasise the need for multimodal approaches that are explicable. Specifically, we propose SHARCS (SHAred Concept Space), a novel concept-based approach to explainable multimodal learning that learns interpretable concepts from different heterogeneous modalities and projects them into a unified concept manifold. An inherent limitation of our approach is that it assumes some relationships between (the same) samples but from different modalities, e.g. multimodal datasets considering different measurements of the same patient. Another possible shortcoming relates to the assumption of the number of discoverable concepts. In some cases, this may lead to many redundant concepts, but this can be mitigated by post-hoc inspection.

We demonstrate the utility and positive benefits of SHARCS, which relate to (i) substantial improvements in generalisation performance compared to other multimodal models, (ii) valuable predictions even when one modality is missing, (iii) high-quality concepts that match expected ground truths, and (iv) enabling insights about the task and other modalities. Moreover, as we discussed in Section 4, in some situations, one modality can be more expressive than the others, thus it can be used for retrieval. This ability can be further extended to intervention tasks, where the model can propose possible interventions applicable to the other modalities. Considering that one of the key characteristics of CBMs is the possibility to interact with a human expert via allowing them to modify the concepts (and as a consequence - the prediction), this capability of SHARCS can enable better quantitative and qualitative analyses, particularly in many real-world safety-critic domains.

We believe that this work could form the basis for the development and analysis of interpretable multimodal approaches.

REPRODUCIBILITY

In Section 2 we introduce our concept-based approach for explainable multimodal learning, and further discuss the implementation details in Appendix A. To ensure the work is readily reproducible, besides descriptions of the experimental setup provided in Section 3 and the supplementary material (Appendices B and C) - we also provide code used for producing the results in the paper.

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

# A SHARCS IMPLEMENTATION DETAILS

## A.1 DIFFERENT CONFIGURATION OF SHARCS

**End-to-end** It is possible to train all SHARCS components simultaneously, allowing a joint optimisation of the task and the concepts found. Therefore, it is also possible to include the loss of the local tasks in Equation 3. However, to use local supervision, we need to implement inside the model $n$ local label predictor function $f_1, \ldots, f_n \in \mathbb{N}$, one for each modality $i = 1, \ldots, n$. The local label predictor function $f_i : C_i \to Y_i$ maps the local concepts from the $i$-th modality to the downstream local task space $Y \subseteq R^{l_i}$, where $l_i$ is the number of classes of the local task of the modality $i$. Therefore the objective function to minimise is the following:

$$\mathcal{L}(\mathbf{y}, \hat{\mathbf{y}}, \mathbf{s}) = \mathcal{T}(\mathbf{y}, \hat{\mathbf{y}}) + \frac{\lambda}{|M|} \sum_{(i,q) \in M \subseteq \binom{\{1,\ldots,n\}}{2}} \left|\left|\mathbf{s}_i - \mathbf{s}_q\right|\right|_2 + \sum_{i=1}^{n} \beta_i \mathcal{T}_i(\mathbf{y_i}, \hat{\mathbf{y_i}}) \tag{7}$$

where $\beta_i \in \mathbb{R}$ is a hyperparameter that controls the strengths of the local loss $\mathcal{T}_i$.

**Sequential** The training process of this method is split in two parts. In the first one, a model similar to Concept Multimodal is trained. Therefore, unimodal models $g_1, \ldots, g_n$ are utilised to compute local concepts, which are concatenated and passed through the label predictor function to solve the downstream task. This part of the entire architecture is trained first, using an objective function equals to $\mathcal{T}$, solving the task using local concepts. Then, the concept encoders functions $g_1, \ldots, g_n$ are frozen. In the second part of the training, local concepts are projected into the shared space by $h_1, \ldots, h_n$, concatenated and used by $f$ to make the final prediction. At this point, the standard loss described in Equation 3 is applied.

**Local pre-training** In this approach, SHARCS' single modality components $g_1, \ldots, g_n$ are trained first, using the same local label predictor functions $f_1, \ldots, f_n$ described in the end-to-end approach to make a prediction. Each is trained using their specific local loss $\mathcal{T}_i$. Then, the concept encoders functions $g_1, \ldots, g_n$ are frozen, while the other SHARCS' modules are employed and trained using the standard objective function described in Equation 3.

## A.2 CONCEPT FINDING ON GRAPH

Although our solution is model agnostic, it is important to treat every modality properly. Therefore, we slightly modify the concept encoder function when it is composed of a Graph Neural Network. Specifically, we applied a modified version of the Concept Encoder Module (CEM)(Magister et al., 2022). In this case, the concept encoder function $g_i$ is composed of a Graph Neural Network $\phi_i : X_i \to H_i$, a Gumbel Softmax (Jang et al., 2017) to find the "node concepts", an add pooling over the nodes of the graph, a batch scaling function and a sigmoid Function. Therefore to find $\mathbf{c}_{im}$, where $i$ is a graph modality, the equation becomes the following:

$$\mathbf{t}_{im} = \phi_i(\mathbf{x}_{im}) \qquad \mathbf{n}_{im} = \sum_{d \in \mathbf{x}_{im}} \sigma(\mathbf{t}_{imd})) \tag{8}$$

$$\mathbf{c}_{im} = \left(1 + \exp\left(-\left(\mathbf{n}_{im} \underset{j \in B_{im}}{\circledast} \mathbf{n}_{ij}\right)\right)\right)^{-1} \tag{9}$$

where $\phi_i$ represents the Graph Neural Network applied to the modality $i$, which outputs the representation of each node $d$ of graph $m$ in the modality $i$, $\sigma$ is the Gumbel Softmax, and $\mathbf{n}$ represents the sum over the node concept of the graph $m$. Therefore, in our solution, the graph concept is related to the occurrences of each node concept.

The issue with CEM is that when it aggregates node concepts, there is no one-to-one mapping between a set of node concepts and graph concepts. This could lead to giving the wrong concept to a graph. Figure 4 shows an example of a situation where two different graphs end up with the same concepts.

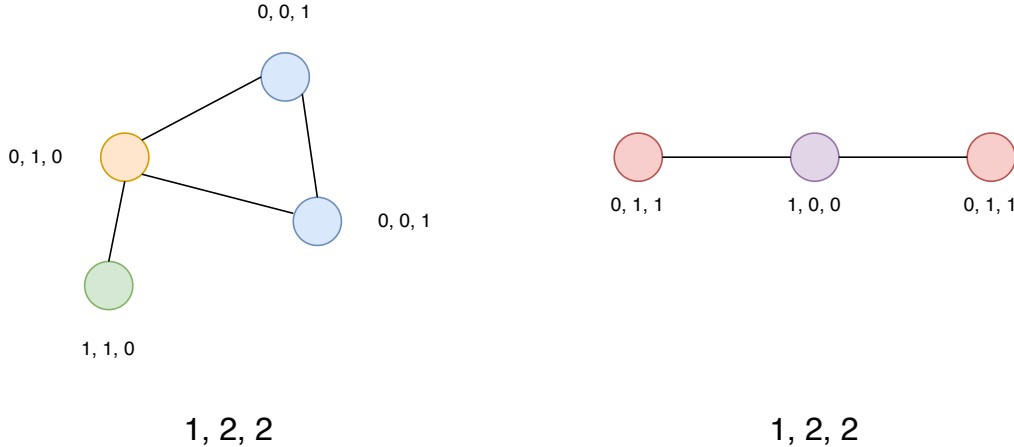

Figure 4: An example of two different graphs with a different set of node concepts described with the same graph concept.

A.3    CODE, LICENCES AND RESOURCES

**Libraries** For our experiments, we implemented all baselines and methods in Python 3.9 and relied upon open-source libraries such as PyTorch 2.0 (Paszke et al., 2019) (BSD license), Pytorch Geometric 2.3 (Fey & Lenssen, 2019) (MIT license) and Sklearn 1.2 (Pedregosa et al., 2011) (BSD license). In addition, we used Matplotlib (Hunter, 2007) 3.7 (BSD license) to produce the plots shown in this paper and Dtreeviz[1] 2.2 (MIT license) to produce the tree visualisations. We will publicly release the code to reproduce all the experiments under an MIT license.

**Resources** We run all the experiments on a cluster with 2x AMD EPYC 7763 64-Core Processor 1.8GHz, 1000 GiB RAM, and 4x NVIDIA A100-SXM-80GB GPUs. We estimate that all the experiments require approximately 50 GPU hours to be completed.

## B    DATASET DETAILS

We design the experiments to understand the potentiality of the proposed solution, described in Section 2. Specifically, We create a synthetic dataset that can validate all the contributions of our method, design two other tasks using existing datasets and use an existing dataset and task to test it in less constrained situations. Each dataset is split into the train (80%) and test set (20%).

### B.1    XOR-AND-XOR

We design a synthetic dataset (XOR-AND-XOR) that contains two modalities: tabular data and graphs. The first contains 6 random bits, but only the first two are meaningful. The second modality contains one of 4 kinds of graphs: (i) 10 nodes which are not connected; (ii) 4 nodes connected in a circle and 6 not; (iii) 6 nodes connected in a circle and 4 not; (iv) 4 nodes connected in a circle, 6 other nodes connected in another circle and the two connected by an edge. They also have a few random edges, and the initial nodes' feature is its betweenness centrality. Each of these graphs can be associated with a combination of the two significant bits of the table, as Figure 5 shows.

In this dataset, there is a local task and a global task. The local one is intra-modality and is the XOR operator between the two meaningful bits or between the above-explained translation from graphs to bits. On the other hand, the global task corresponds to the AND operator between the result of the local tasks. The global task cannot be solved using just one modality, so both pieces of information are needed to classify each entry correctly. However, on this dataset, we do not make supervision on the local task available, letting the model understand it.

---
[1]https://github.com/parrt/dtreeviz

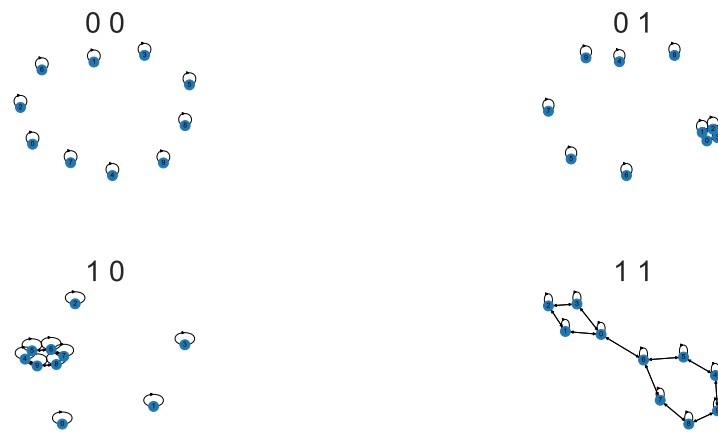

Figure 5: Examples of the conversion from the four main families of graphs to the meaningful bits of the tabular data in the XOR-AND-XOR dataset. In the dataset, they have some additional random edges.

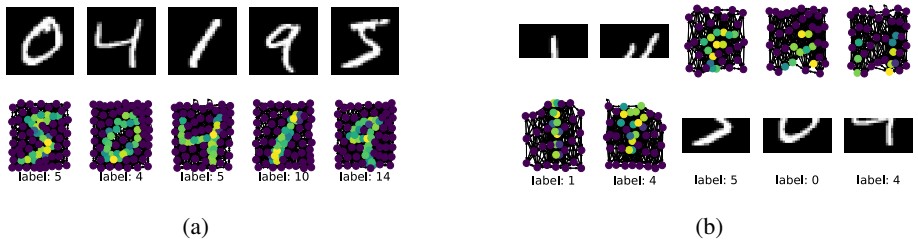

Figure 6: (a) Examples from the MNIST+Superpixels dataset. The shown label is related to the task, which is the sum of the two digits. (b) Examples from the HalfMNIST dataset. The shown label is related to the task, which is the digit represented by joining both parts. Each half can be represented with one of the two modalities.

The entire dataset contains 1000 samples for each modality, the translations from one modality to the other for both modalities and the labels relative to the global task.

## B.2 MNIST + SUPERPIXELS

The second dataset (MNIST+Superpixels) consists in predicting the sum of two digits described in two different ways. One is in the shape of an MNIST image (Deng, 2012), and the other is represented as a superpixel graph of another MNIST image (Monti et al., 2016). Here the local task is correctly classifying the single digit, while the global is predicting the sum of the two. Figure 6a shows five samples from this dataset, including the global label.

In this task, local supervision is available. Therefore, the dataset contains 60000 couple of digits, both described as a graph and as an image (during training, we use the graph of digit 1 and the image of digit 2), the labels for the local tasks and the ones for the global task.

## B.3 HALF-MNIST

The third experiment uses another dataset containing the same modalities as the previous one (MNIST and MNIST Superpixels) but differently. In this case, the task is to predict the single digit, but one half is in the shape of an image, and the other half is described as a superpixel graph. It is important



Figure 7: Examples from the CLEVR dataset, where there is a text caption and an image of an object. The label is True if the caption correctly describes the image, otherwise is False.

to say that some of the upper halves are images, some are graphs, and the same last for the bottom halves. Figure 6b shows five samples from this dataset.

Here, the global task is the same as the local one, but it is possible to use more information from a different modality to solve it. Therefore, this dataset contains 60000 digits described as graphs and images (only half image and half graph are used during training) and labels for the global task.

### B.4 CLEVR

The last dataset used is a version of CLEVR we generate using the official repository. [2] Our version is inspired by Saqur & Narasimhan (2020), where each image contains one object and the relative caption can match or not the image. Specifically, in our case, every caption contains four attributes used to describe the scene in the image. They are the shape (sphere, cylinder, cube), the size (big, small), the colour (green, yellow, gray, red, purple, cyan, blue, brown) and the material (matte, metallic) of the object. The task is to predict if the caption correctly describes the image. If the label is equal to True, we consider that a connection between the two modalities, as it means that the two modalities represent the same object. Figure 7 shows five samples taken from this dataset, the top row represents the captions, while the bottom is about the images.

In this situation, there is no local task. Therefore, this dataset contains 8320 couple of captions and images, their translation in the other modality and the labels.

## C MODELS DETAILS

In this section, we describe in detail the configuration of SHARCS used in each experiment. Then, we add only the missing or different information needed to build the other models used, as most of the details are in common between our solutions and baselines.

In general, single modality models used only the DL model inside of the respective $g_i$, with (or without) a sigmoid function, if it is a concept-based (concept-less) solution. Simple Multimodal and Relative representation solutions employ the DL models inside $g_i$ and the label predictor $f$, while Concept Multimodal also uses batch scaling and the sigmoid inside $g_i$.

### C.1 XOR-AND-XOR

On this task, we trained SHARCS with the end-to-end configuration, as we do not have local supervision. It is composed of two $g_i$ concept encoder functions, one for each modality. To handle the

---

[2]https://github.com/facebookresearch/clevr-dataset-gen

graph modality, the DL model inside of $g_1$ is composed of 5 layers of Graph Convolutional Networks (Kipf & Welling, 2016) with LeakyReLU as the activation function. The input size is 1 as described in Appendix B.1, the hidden size of all the intermediate layers is 30, while the output dimension of $g_1$ is 7. On the other hand, a simple 2-layer MLP with a ReLU as the activation function is the DL model of $g_2$, which takes tabular data as input. The input size is 8, the hidden size is 30, and the output dimension is equal to 7. SHARCS uses Batch Normalisation as batch scaling and Sigmoid to compute concepts, but on the graph modality follows the approach described in Appendix A.2. The second set of concept encoders $h_1$ and $h_2$ are 2-layer MLPs with a ReLU as the activation function, with an input dimension of 8, as well as the hidden and output size. Finally, the label prediction function $f$ is a 2-layer MLP with a ReLU as the activation function, with an input dimension of 16, a hidden size of 10 and an output dimension equals to the number of classes, which is 2.

An additional detail for single modality models is their label prediction function $f_i$, one for each modality, which is a 2-layers MLPs with a ReLU as the activation function, with an input dimension of 8, a hidden size of 10 and an output dimension of 2.

In terms of learning process, we used a Binary Cross Entropy Loss (BCELoss) with Logits (which incorporates a sigmoid layer before computing the BCELoss) as $\mathcal{T}$, a $\lambda$ equals to 0.1, and at every iteration, we took 10% of randomly draw samples to compute the distance. Other hyperparameters used to train the models are the Batch Size used (64), the number of epochs (150) and the Learning Rate used by an Adam optimizer (0.001). However, we train the unimodality models of Relative representation models for 150 epochs and its label predictor function for other 150 epochs.

## C.2 MNIST+SUPERPIXELS AND HALFMNIST

On MNIST+Superpixels and HalfMNIST, we used an almost identical setup. We trained SHARCS with the local pre-training configuration, as we have local supervision. It is composed of two $g_i$ concept encoder functions, one for each modality. To handle the graph modality, the DL model inside of $g_1$ is composed of 2 layers of SplineCNN (Fey et al., 2018) with ELU as the activation function, similar to the SplineCNN model described in the original paper. Therefore, a max pooling operator based on the Graclus method (Dhillon et al., 2007) is applied after every layer. The input size is 1, the hidden size of all the intermediate layers is 32, and the output dimension of $g_1$ is 12. On the other hand, a Convolutional Neural Network is the DL model of $g_2$. It is composed of the following layers: a Convolutional Layer (input channel=1, output channel=16, kernel size=5, padding=2, stride=1), a ReLU, a MaxPool with a kernel size of 2, a Convolutional Layer (input channel=16, output channel=16, kernel size=5, padding=2, stride=1), a ReLU, a MaxPool with a kernel size of 2, then the output is flattened and taken as input from a 2-layer MLP with a ReLU as the activation function, with a hidden dimension of 64 and output size of 12. Moreover, SHARCS uses Batch Normalisation as batch scaling and sigmoid to compute concepts, but on the graph modality follows the approach described in Appendix A.2. The second set of concept encoders $h_1$ and $h_2$ are 2-layer MLPs with a ReLU as the activation function, with an input dimension of 12, as well as the output size and a hidden size of 64. Finally, the label prediction function $f$ is a 2-layer MLP with a ReLU as the activation function, with an input dimension of 24, a hidden size of 128 and an output dimension equals to the number of classes, which is 19 for MNIST+Superpixels and 10 for HalfMNIST. As we apply the local pre-training configuration, in the first part of the training, we used some local label predictor function $f_i$, one for each modality. They are 2-layer MLPs with a ReLU as the activation function, with an input dimension of 12, a hidden size of 64 and an output dimension equals to the number of classes of the local task, which is 10 for both datasets. Other unimodal baselines also use these local label predictor functions.

Regarding the learning process, we used a BCELoss with Logits both with local and global tasks, a $\lambda$ equals to 0.1, and at every iteration, we took 10% of randomly drawn samples to compute the distance. Other hyperparameters used to train the models are the Batch Size used (64), the number of epochs used to pretrain the unimodal models (15) and the additional epochs used to train the second part of SHARCS (15). The learning rate used by the Adam optimiser is equal to 0.01 for the Graph Neural Network and 0.001 for all the other layers of the model. However, we train the unimodality models of Relative representation models for 15 epochs and its label predictor function for other 20 epochs.

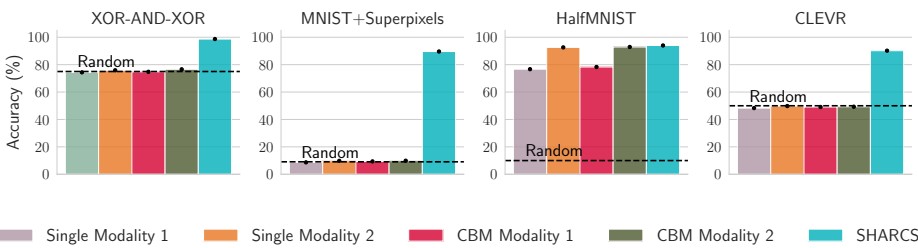

Figure 8: Accuracy of unimodal models and SHARCS on all datasets. SHARCS outperforms all the other models on all tasks.

## C.3 CLEVR

On this task, we trained SHARCS with the sequential configuration, as we do not have local supervision and want to discover local concepts that are not influenced by the other modality. It is composed of two $g_i$ concept encoder functions, one for each modality. To handle the image modality, the DL model inside of $g_1$ is a pretreated ResNet18 (He et al., 2015), followed by a Dense layer that reduced the output size of the ResNet to 24. On the other hand, a simple 2-layer MLP with a ReLU as the activation function is the DL model of $g_2$, which takes the TF-IDF representation of the caption received as input. The input size is 22, the hidden size is 48, and the output dimension is equal to 24. SHARCS uses Batch Normalisation as batch scaling and sigmoid to compute concepts. The second set of concept encoders $h_1$ and $h_2$ are 2-layer MLPs with a ReLU as the activation function, with an input dimension of 24, as well as the hidden and output size. Finally, the label prediction function $f$ is a 2-layer MLP with a ReLU as the activation function, with an input dimension of 48, a hidden size of 10 and an output dimension equals to the number of classes, which is 2.

An additional detail for single modality models is their label prediction function $f_i$, one for each modality, which is a 2-layers MLPs with a ReLU as the activation function, with an input dimension of 24, a hidden size of 24 and an output dimension of 2.

In terms of learning process, we used a BCELoss with Logits, a $\lambda$ equals to 0.1, and at every iteration, we took the samples with the label equals to True out of 20% of randomly drawn samples to compute the distance. Other hyperparameters used to train the models are the Batch Size used (64), the number of epochs used by all models and in the first part of the training of SHARCS (30), the additional epochs used in the second part of the training of SHARCS (20) and the Learning Rate used by an Adam optimizer (0.001). In addition, we train the unimodality models of Relative representation models for 30 epochs and its label predictor function for other 20 epochs.

## D  ADDITIONAL RESULTS

This section includes additional results and consideration of the experiments presented in Section 3.

**Broader Impacts** We do not believe this approach can have a direct harmful impact when applied in AI systems. On the contrary, it can positively influence the development of models for safety-critical domains, such as healthcare.

**Detailed results of experiments**

Figure 8 shows the performance of unimodal models compared to SHARCS in all tasks. It is clear how 3 out of the 4 datasets we designed are not solvable by unimodal models, proving our design choice. Furthermore, Table 4 shows the Accuracy for all the models trained and the Completeness Score for the multimodal interpretable models. It gives more detailed results and compares together all the trained models. On the other hand, Table 5, shows the result of an analysis we performed on CLEVR, where we checked for each model which characteristics of the retrieved sam- ple matched with the ones of the object used as the source.

**Interpretability** We present the visual results for each dataset to give a better idea of the performance of our solution. We show the retrieved examples per modality in each dataset, the learnt shared

Table 4: Accuracy (%) and Completeness Score (%) of SHARCS compared to non-interpretable unimodal models (Simple Modality 1 and Simple Modality 2), non-interpretable multimodal models (Simple Multimodal and Relative representation), interpretable unimodal models (CBM Modality 1 and CBM Modality 2) and interpretable multimodal baselines (Concept Multimodal). Generally, SHARCS achieves better (or comparable) performance than the other baselines, producing better and more compact concepts.

| Model | XOR-AND-XOR | | MNIST+SuperP. | | HalfMNIST | | CLEVR | |
|---|---|---|---|---|---|---|---|---|
| | Acc. | Compl. | Acc. | Compl. | Acc. | Compl. | Acc. | Compl. |
| Mod 1 | $74.4 \pm 0.7$ | - | $8.7 \pm 0.1$ | - | $76.7 \pm 0.2$ | - | $48.3 \pm 0.3$ | - |
| Mod 2 | $75.9 \pm 1.4$ | - | $9.8 \pm 0.1$ | - | $92.6 \pm 0.2$ | - | $49.8 \pm 0.1$ | - |
| CBM 1 | $74.8 \pm 0.0$ | - | $9.4 \pm 0.1$ | - | $78.3 \pm 0.1$ | - | $49.1 \pm 0.5$ | - |
| CBM 2 | $76.6 \pm 1.3$ | - | $9.9 \pm 0.2$ | - | $92.9 \pm 0.1$ | - | $49.3 \pm 0.4$ | - |
| Simple | $99.3 \pm 0.5$ | - | $86.6 \pm 3.0$ | - | $94.2 \pm 0.2$ | - | $59.5 \pm 9.5$ | - |
| Concept | $99.0 \pm 0.8$ | $96.2 \pm 1.2$ | $88.2 \pm 0.1$ | $78.9 \pm 1.4$ | $93.9 \pm 0.0$ | $91.3 \pm 0.1$ | $90.1 \pm 1.0$ | $\mathbf{82.3 \pm 1.2}$ |
| Relative | $\mathbf{99.5 \pm 0.3}$ | - | $80.4 \pm 0.2$ | - | $\mathbf{95.6 \pm 0.1}$ | - | $48.7 \pm 0.5$ | - |
| SHARCS | $98.7 \pm 0.5$ | $\mathbf{98.0 \pm 1.2}$ | $\mathbf{89.6 \pm 0.1}$ | $\mathbf{88.7 \pm 0.2}$ | $94.0 \pm 0.1$ | $\mathbf{92.6 \pm 0.3}$ | $\mathbf{90.2 \pm 0.2}$ | $81.5 \pm 1.1$ |

Table 5: Accuracy (%) of Relative representation, Concept Multimodal and SHARCS in retrieving a specific characteristic in a modality using the other. SHARCS attains higher figures than other models on every characteristic.

| Model | Modality | Shape | Size | Material | Color | Mean |
|---|---|---|---|---|---|---|
| Concept | Text | $31.7 \pm 2.3$ | $46.0 \pm 3.0$ | $52.1 \pm 0.2$ | $16.2 \pm 3.7$ | $36.5 \pm 0.6$ |
| | Image | $30.0 \pm 0.2$ | $45.4 \pm 5.3$ | $51.3 \pm 2.6$ | $10.8 \pm 1.3$ | $34.3 \pm 0.6$ |
| Relative | Text | $29.9 \pm 1.0$ | $50.5 \pm 0.7$ | $50.0 \pm 0.6$ | $13.3 \pm 1.2$ | $35.9 \pm 0.3$ |
| | Image | $33.0 \pm 1.0$ | $49.6 \pm 0.2$ | $49.0 \pm 1.0$ | $11.1 \pm 0.7$ | $35.6 \pm 0.2$ |
| SHARCS | Text | $\mathbf{56.8 \pm 1.4}$ | $\mathbf{63.6 \pm 3.5}$ | $\mathbf{53.9 \pm 1.8}$ | $\mathbf{30.2 \pm 5.6}$ | $\mathbf{51.1 \pm 2.0}$ |
| | Image | $\mathbf{51.4 \pm 2.4}$ | $\mathbf{61.5 \pm 1.7}$ | $\mathbf{53.4 \pm 2.6}$ | $\mathbf{27.5 \pm 4.0}$ | $\mathbf{48.5 \pm 1.9}$ |

space and the decision tree. The results presented here for the same dataset included in Section 3 are run with a different random seed to show how solid the performance of SHARCS is. Figure 9 shows the retrieved examples by SHARCS, Relative Representation and Concept Multimodal in the XOR-AND-XOR dataset. In particular, in Figure 9a, it is interesting to see how SHARCS retrieves tabular data that are not constrained to be closer but have the same semantic meaning for the local task, which is False in the XOR operator. Figure 10 illustrates the same experiments with MNIST+Superpixels and Figure 11 with HalfMNIST. Finally, Figure 12 shows the retrieval capability of these models on the CLEVR dataset. In all these experiments, it can be seen that the quality of the retrieved examples is higher than the others, where the Relative Representation is not always accurate and Concept Multimodal resembles random retrieval. The second set of images visually confronts the shared space learnt by SHARCS and Concept Multimodal. For this purpose, we visualise the tSNE representation of the shared concepts for SHARCS and the local concepts for Concept Multimodal. Figure 13 shows these shared spaces for the MNIST+Superpixels dataset, Figure 14 for HalfMNIST and Figure 15 for CLEVR. It is clear how the concept representation learnt by SHARCS for one modality overlaps with that for the other, especially when considering semantically similar examples from different modalities that are closer in the space representation. All these results are expected by design since we force the model to produce the shared space with these properties. Finally, part of the decision trees used to compute the completeness score is visualised. At every split, it shows the concept that is considered to make the decision, and it can be active (right branch) or non-active (left branch). If the node is not the roof, it also shows three samples with the highest (concept active) or the lowest (concept non-active) value for the concepts of the previous split, among the ones that respect all the previous split conditions. Each leaf shows the class distribution of the samples that it represents, in addition to the most characteristic samples. Moreover, the root of the tree uses the most influential concept for the classification task, as it is by definition the one that brings the highest Information Gain, and the same is applicable to the following splits. For example, Figure 16 shows the decision tree used in the XOR-AND-XOR dataset. Specifically, you can see that if Concept 11 is active, the prediction is always Class 0 (False). As you can see, if a sample has Concept 11 active, it means that it has both tabular significant digits equal to 1, which implies that the local XOR

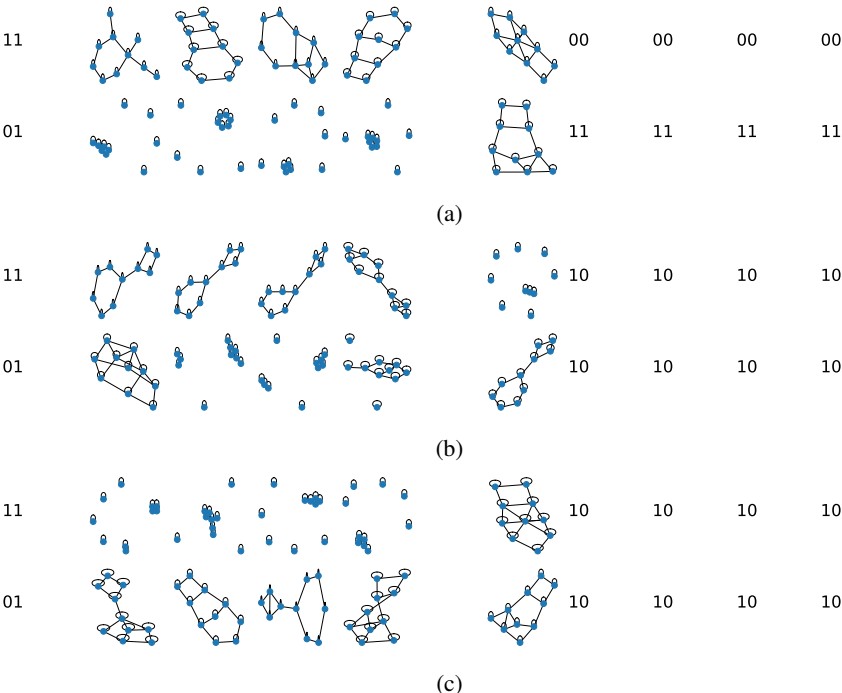

Figure 9: Retrieval examples obtained by (a) SHARCS, (b) Relative representation, and (c) Concept Multimodal on the XOR-AND-XOR dataset. The top two rows are samples of retrieved graphs using tabular data, while the bottom two are retrieved tabular entries using graph samples.

operation is False and as a consequence the global AND operation is False, no matter what is the other modality. Furthermore, the following split is focused on Concept 4, which is curiously the corresponding concept in the shared space of the graph modality for Concept 11 (7 is the number of concepts per modality, so 11 - 7 = 4). This split represents the same underlying idea of the previous one but for the graph modality. If the concept is active, it means that the graph is connected (False in the local XOR operation). Therefore, it shows also how the concepts from one modality are related and translated into the other, confirming that the concept shared space created is meaningful. Finally, Figure 17 shows part of the first three layers of the Decision tree used in CLEVR.

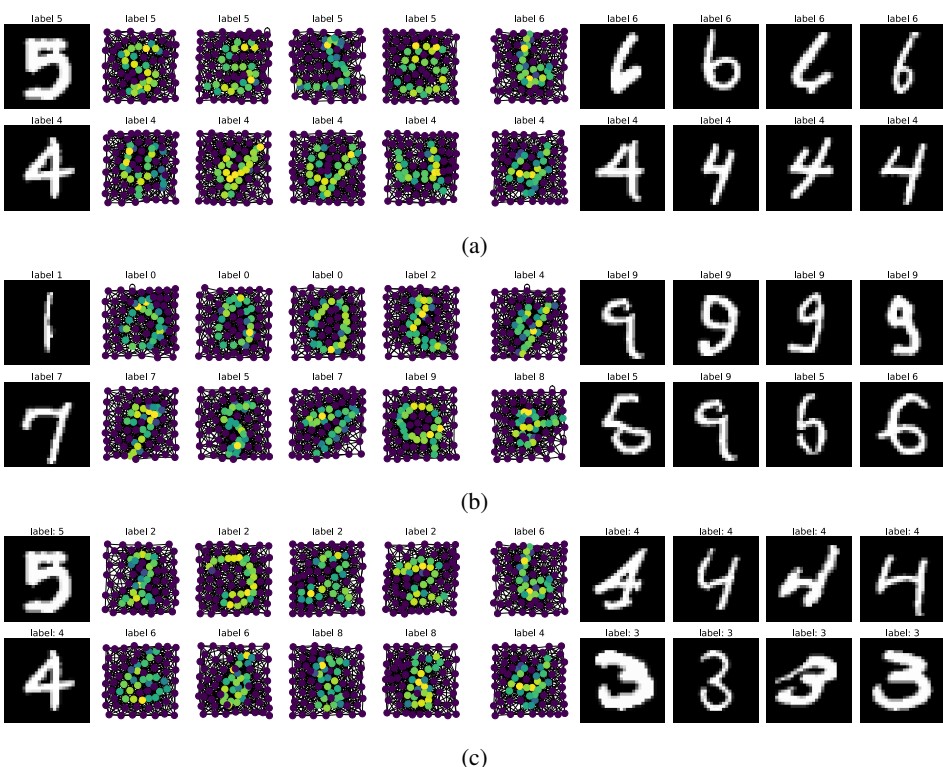

Figure 10: Retrieval examples obtained by (a) SHARCS, (b) Relative representation, and (c) Concept Multimodal on the MNIST+Superpixels dataset. The top two rows are samples of retrieved graphs using images, while the bottom two are retrieved images using graph samples.

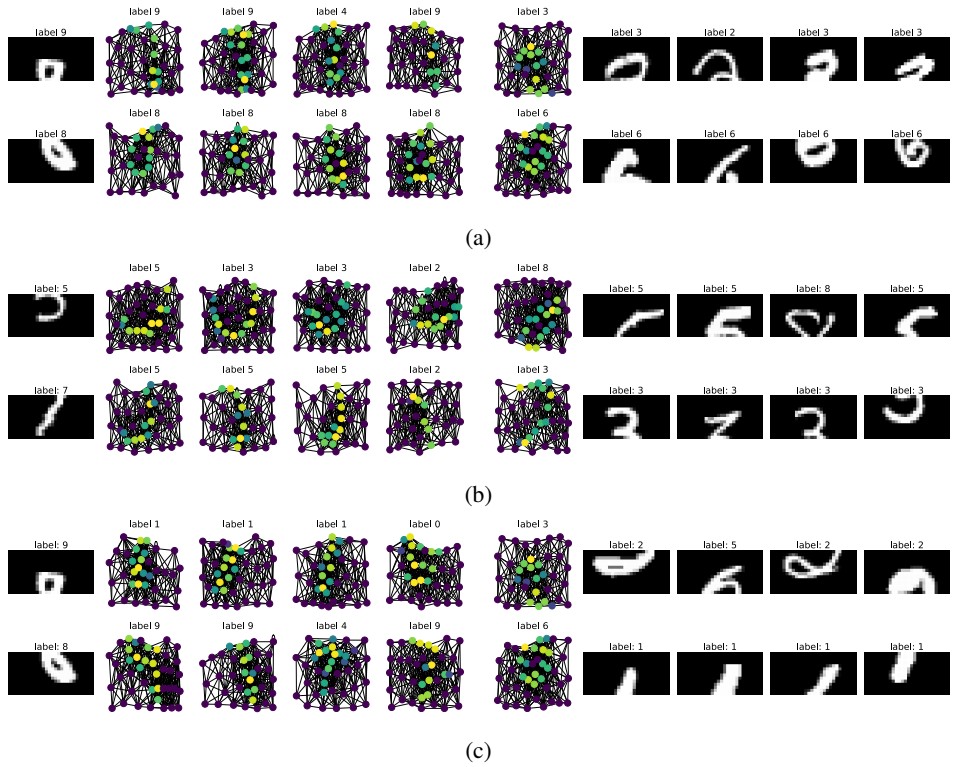

Figure 11: Retrieval examples obtained by (a) SHARCS, (b) Relative representation, and (c) Concept Multimodal on the MNIST+Superpixels dataset. The top two rows are samples of retrieved graphs using images, while the bottom two are retrieved images using graph samples.

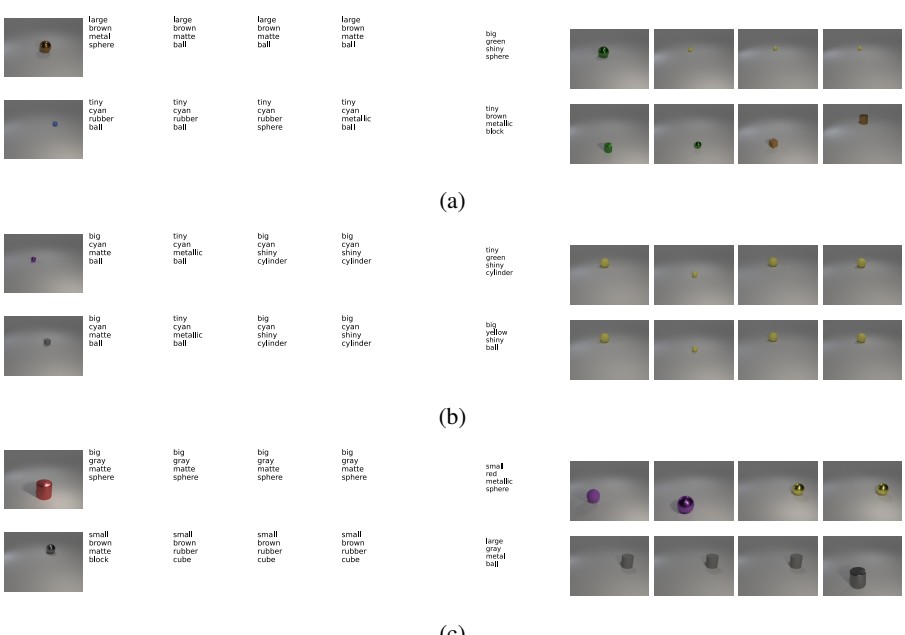

Figure 12: Retrieval examples obtained by (a) SHARCS, (b) Relative representation, and (c) Concept Multimodal on the CLEVR dataset. The top two rows are samples of retrieved text using images, while the bottom two are retrieved images using graph samples.

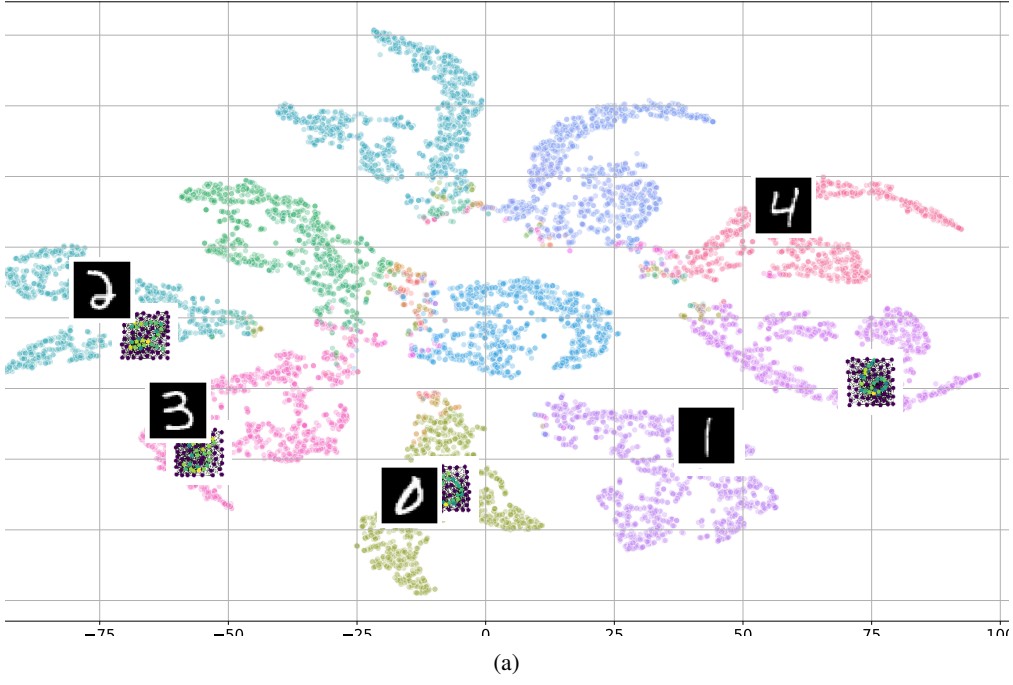

(a)

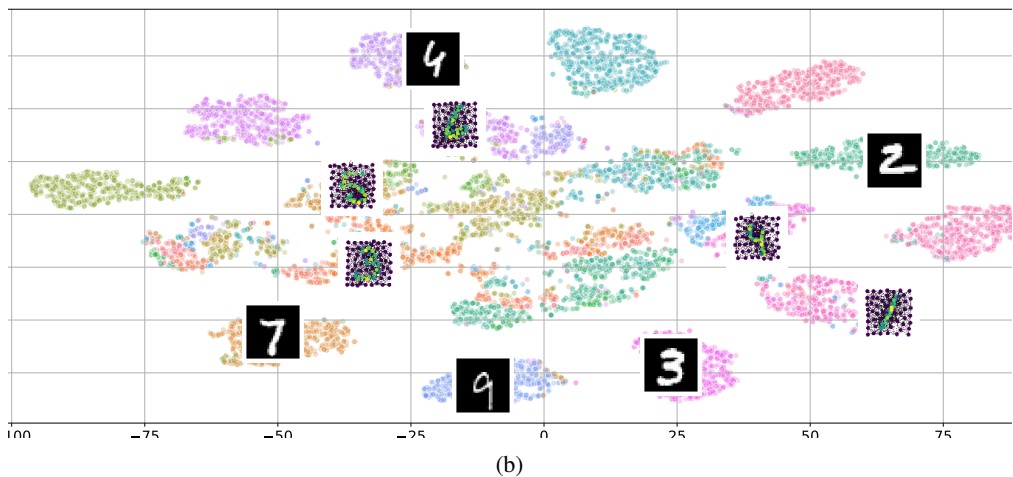

(b)

Figure 13: tSNE plot of the concept space. The images represent the centroid of the top-5 common concepts per modality in the MNIST+Superpixels dataset (a) SHARCS (b) Concept Multimodal

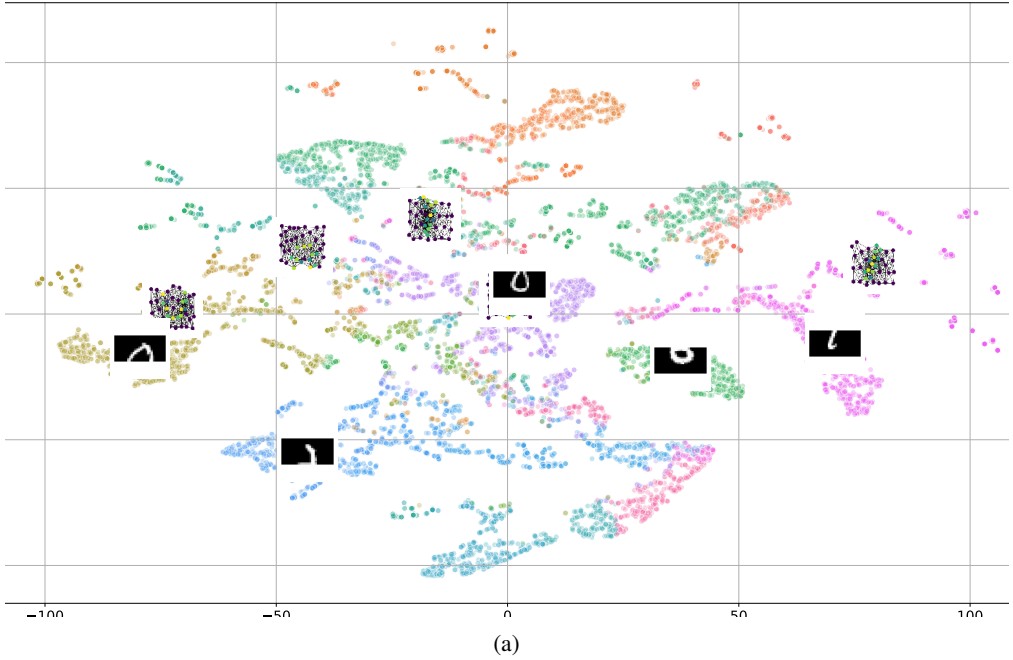

(a)

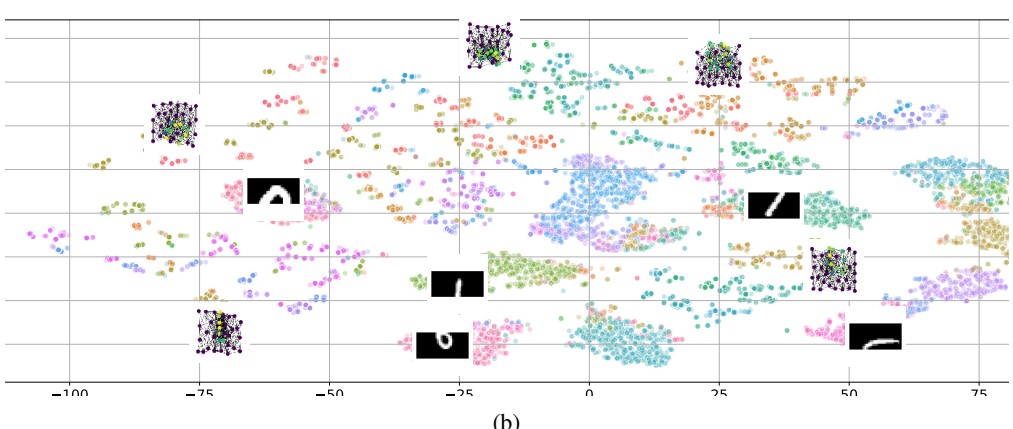

(b)

Figure 14: tSNE plot of the concept space. The images represent the centroid of the top-5 common concepts per modality in the HalfMNIST dataset (a) SHARCS (b) Concept Multimodal

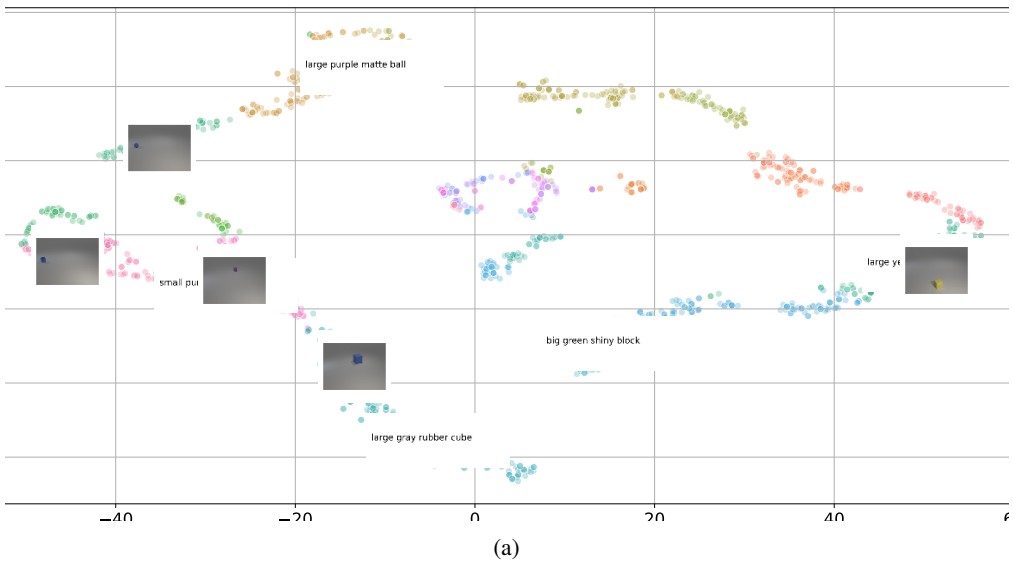

(a)

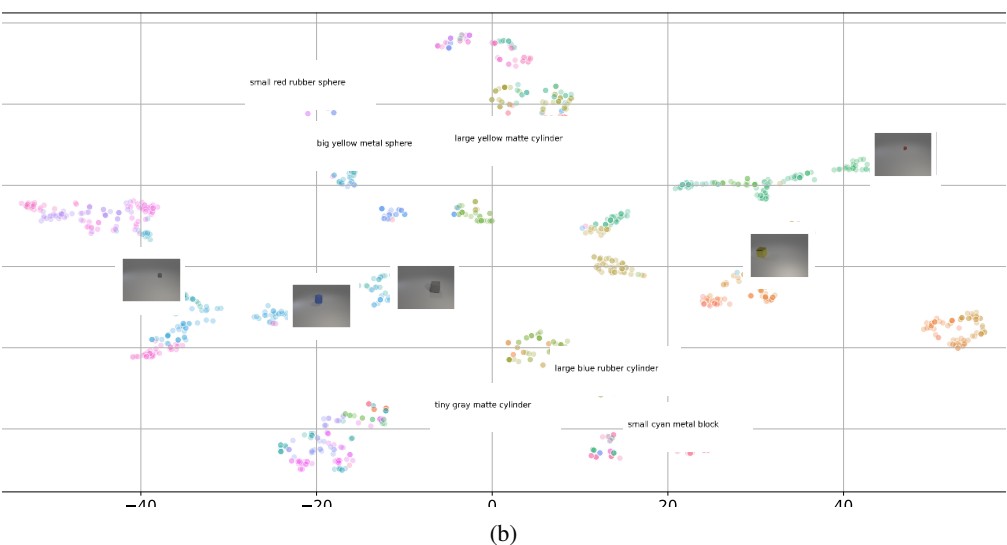

(b)

Figure 15: tSNE plot of the concept space. The images represent the centroid of the top-5 common concepts per modality in the CLEVR dataset (a) SHARCS (b) Concept Multimodal

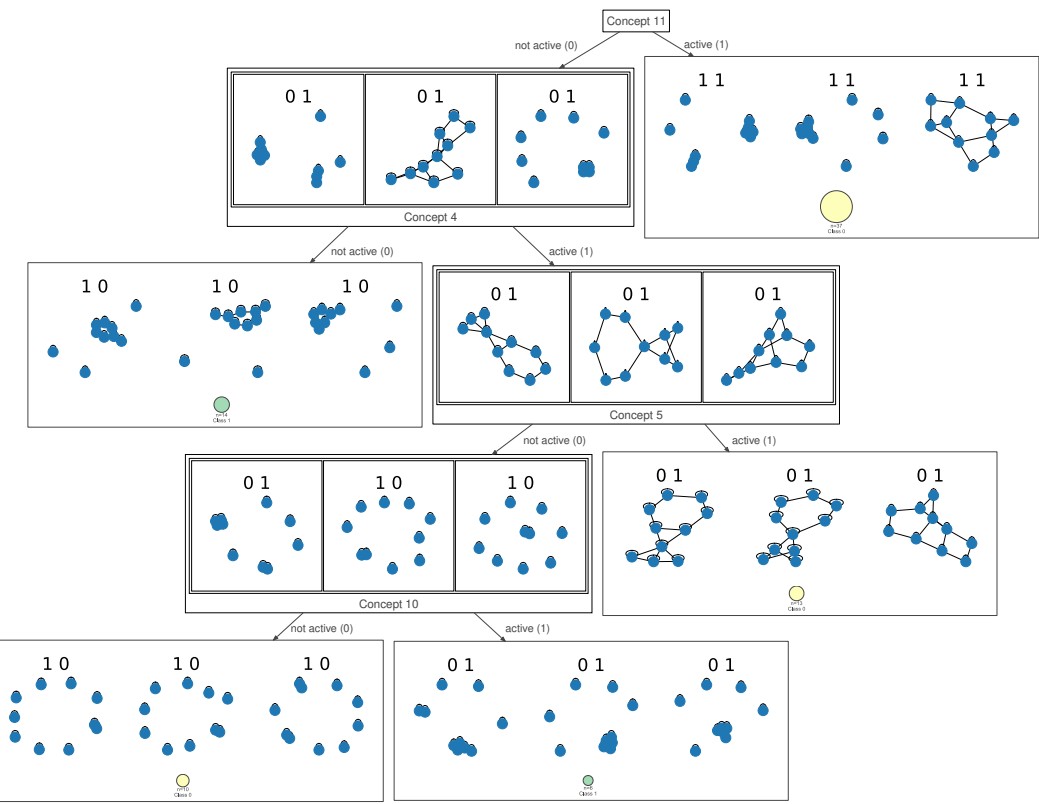

Figure 16: Decision tree visualisation of SHARCS concepts on the XOR-AND-XOR dataset. Every split shows the combined concept closer to the cluster's centroid lower and greater than the splitting criteria. In addition, each leaf shows the class distribution of the samples that it represents.

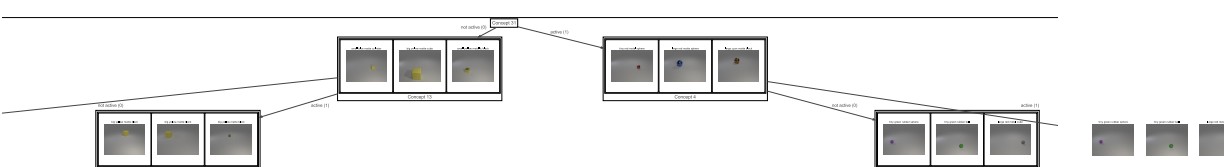

Figure 17: Visaulisation of part of the first 3 layers of a Decision tree trained on SHARCS concepts on the CLEVR dataset. Every split shows the combined concept closer to the cluster's centroid lower and greater than the splitting criteria. In addition, each leaf shows the class distribution of the samples that it represents.

