# OpenReview forum: "SHARCS: SHARed Concept Space for\\Explainable Multimodal Learning"
_ICLR.cc/2024/Conference — Submitted to ICLR 2024_

### Official Review · Reviewer_yLh6 · 2023-10-24

**Soundness:** 3 good
**Presentation:** 2 fair
**Contribution:** 2 fair
**Rating:** 5
**Confidence:** 4

**Summary:**

This paper proposes a concept-based approach for explainable multimodal learning named SHARCS. It projects the local concept of each modality into a shared concept space and then concatenates concepts from different modalities for prediction. SHARCS indicates the potential connection between modalities for interpretability and also shows higher performance on multimodal tasks.

**Strengths:**

1. This paper focuses on a key problem of the interpretability of multimodal learning. There are few multimodal learning methods that provide explainable concepts before.
2. The proposed method is novel enough and technically reasonable. This paper describes the method logically and in detail.
3. This paper provides sufficient implementation details for reproducibility.

**Weaknesses:**

1.	The experimental results provided are not enough to verify the performance and interpretability of SHARCS. (1) There is a lack of quantitative evaluation for the interpretability of SHARCS, especially when interpretability is the main claim of this paper. (2) The experiments are conducted on simple datasets like MNIST and CLEVR. It is better to discuss the possibility of experiments on more complex and real-world multimodal datasets like COCO-caption.

2.	The organization of this paper is not reasonable enough. The evaluation metrics should be one important part but the definition is not clear, especially the completeness score and the decision tree. While the learning process section is slightly abundant, which can be moved to the appendix.

3.	There are some grammatical mistakes and typos in the text and the appendix, please carefully proofread the whole paper.

**Questions:**

1.	Is it possible to quantitatively evaluate the interpretability of SHARCS and compare it with other methods?
2.	Is the completeness score a suitable and complete metric for interpretability? Can SHARCS serve as a baseline for future research?

---

### Official Review · Reviewer_D8iH · 2023-10-31

**Soundness:** 2 fair
**Presentation:** 3 good
**Contribution:** 3 good
**Rating:** 6
**Confidence:** 3

**Summary:**

This paper introduces SHARCS, a model-agnostic concept-based approach for explainable multimodal learning. The proposed method learns and maps interpretable concepts from different heterogeneous modalities into a single unified concept-manifold, leading to an intuitive projection of semantically similar cross-modal concepts.  The authors carried out experiments on four datasets, demonstrating the superior performance of SHARCS over a range of unimodal and multimodal baseline models.

**Strengths:**

1. SHARCS can learn concepts from different heterogeneous modalities and project them into a unified concept manifold, which enables comprehensible explanations at different levels (modality-specific or global) and between different modalities.
2. Its ability to handle scenarios with absent modalities renders the approach particularly pragmatic in real-world applications, where the presence of missing modalities is quite common.
3. Thorough ablation study and detailed case analyses. The paper is well-written and easy to follow.

**Weaknesses:**

1. The performance data presented in Table 1 appears to be underwhelming and may raise concerns about the effectiveness of the proposed approach.
2. The paper falls short in its analysis and comparison of contemporary multimodal explanation techniques [a][b]. It notably omits a discussion on the applicability of the approach with respect to recent multimodal foundation models like CLIP, OFA, and LLaVA, leaving uncertainties regarding its adaptability.
3. As noted within the paper, the necessity for post-hoc inspection to address redundant concepts could be seen as a limitation, potentially hindering the usability.

[a] Y. Liu and T. Tuytelaars, ‘‘A deep multi-modal explanation model for zero-shot learning,’’ IEEE Trans. Image Process.
[b] A Review on Explainability in Multimodal Deep Neural Nets, GARGI JOSHI etc.

**Questions:**

1. Can this approach work with multimodal embedding models such as CLIP, and generative models such as LLaVA?
2. Can it handle cases where multiple modalities are missing?

---

### Official Review · Reviewer_w4Zt · 2023-11-01

**Soundness:** 2 fair
**Presentation:** 3 good
**Contribution:** 2 fair
**Rating:** 3
**Confidence:** 4

**Summary:**

This paper proposes a method for explaining multimodal deep learning methods called SHARCS (SHARed Concept Space). SHARCS learns and maps interpretable concepts from different modalities into a single unified concept-manifold, which leads to an intuitive projection of semantically similar cross-modal concepts. The authors use this approach for explaining task predictions, improving downstream predictive performance, retrieval of missing modalities, and cross-modal explanations.

**Strengths:**

1. The problem of explaining multimodal models is certainly important.
2. The idea makes sense and is well explained.
3. The paper is generally well-written.
4. There are some interesting controlled multimodal datasets that test specific multimodal interactions.

**Weaknesses:**

1. Tables 1 and 2 need a lot more baselines, including entire fields of study in multimodal fusion, multimodal factorized models, multimodal contrastive learning... right now there is only 'Simple Multimodal,’ which combines uninterpretable embedded representations from individual local models (I'm assuming this is some encoders -> concatenate fusion -> classifier), 'Concept Multimodal,’ which does late fusion? and 'Relative Representations’ citing (Moschella et al., 2023). I'm not sure what these baselines are, their motivation, and why they were chosen, but they ignore so much work in the broader multimodal community. The authors should include them from and refer to https://arxiv.org/abs/2209.03430 for an extensive review.

2. Section '4.2 INTERPRETABILITY' contains a lot of claims that are not evaluated.

---- SHARCS discovers meaningful concepts -> there are only several qualitative anecdotes for this. Can you define meaningful, perhaps with a human annotator or human-in-the-loop evaluation? Are real humans able to annotate the concepts given SHARCS visualizations? Eg. see how MultiViz https://arxiv.org/abs/2207.00056 does it. There should be discussion and comparison to MultiViz https://arxiv.org/abs/2207.00056 and the referenced papers within it on interpreting multimodal models.

---- SHARCS concepts shed light on how the task can be solved -> same thing, can you actually give SHARCS concepts to a human and see if the human can solve the task?

---- SHARCS explains one modality using the other -> needs rigorous evaluation on a retrieval benchmark, of which there are many estalished ones.

3. Are there any computational difficulties of the method?

4. Can you show or describe a sample user interface of the visualization outputs from SHARCS, and how you would envision presenting them to users of multimodal models? This would be a useful addition to the paper.

5. How can you evaluate if SHARCS has successfully learned the right concepts? There should be some sanity checks and evaluation for this.

**Questions:**

see weaknesses above

---

### Meta-Review · Area_Chair_7jzT · 2023-12-06

**Metareview:**

This paper addresses explainability and interpretability in multimodal deep learning by mapping concepts from various modalities into a single unified space. This mapping is then used to perform a projection of semantically close cross-modal concepts. The biggest criticism from all the reviewers is about experimentation. And according to the reviewers it's not clear if the objective of the paper was achieved.

**Justification For Why Not Higher Score:**

The experiments are not convincing and it's not clear if the paper achieved the objectives.

**Justification For Why Not Lower Score:**

N/A

---

### Decision · Program_Chairs · 2024-01-16

Reject